# Image Perceptual Similarity Metrics for the Assessment of Basal Cell Carcinoma

**DOI:** 10.3390/cancers15143539

**Published:** 2023-07-08

**Authors:** Panagiota Spyridonos, Georgios Gaitanis, Aristidis Likas, Konstantinos Seretis, Vasileios Moschovos, Laurence Feldmeyer, Kristine Heidemeyer, Athanasia Zampeta, Ioannis D. Bassukas

**Affiliations:** 1Department of Medical Physics, Faculty of Medicine, School of Health Sciences, University of Ioannina, 45110 Ioannina, Greece; 2Department of Skin and Venereal Diseases, Faculty of Medicine, School of Health Sciences, University of Ioannina, 45110 Ioannina, Greece; ggaitan@uoi.gr (G.G.); athanasiazampeta@gmail.com (A.Z.); ibassuka@uoi.gr (I.D.B.); 3Department of Computer Science & Engineering, School of Engineering, University of Ioannina, 45110 Ioannina, Greece; arly@cs.uoi.gr; 4Department of Plastic Surgery and Burns, Faculty of Medicine, School of Health Sciences, University of Ioannina, 45110 Ioannina, Greece; drseretis@uoi.gr (K.S.); billmosh@icloud.com (V.M.); 5Department of Dermatology, Inselspital, Bern University Hospital, University of Bern, 3010 Bern, Switzerland; laurence.feldmeyer@insel.ch (L.F.); kristine.heidemeyer@insel.ch (K.H.)

**Keywords:** basal cell carcinoma, scar assessment, perceptual similarity, texture similarity, color similarity, convolutional neural network

## Abstract

**Simple Summary:**

The impact of basal cell carcinomas (BCCs) on a patient’s appearance can be significant. Reliable assessments are crucial for the effective management and evaluation of therapeutic interventions. Given that color and texture are critical attributes that describe the clinical aspect of skin lesions, our focus was to devise metrics that capture the way experts perceive deviations of target BCC areas from the surrounding healthy skin. Using computerized image analysis, we explored various similarity metrics to predict perceptual similarity, including different color spaces and distances between features from image embeddings derived from a pre-trained deep convolutional neural network. The results are promising in providing a valid, reliable, and affordable modality, enabling more accurate and standardized assessments of BCC tumors and post-treatment scars. Our approach to modeling color and texture lesion similarity from the surrounding healthy skin is a promising paradigm for the further development of a valid and reliable scar assessment tool.

**Abstract:**

Efficient management of basal cell carcinomas (BCC) requires reliable assessments of both tumors and post-treatment scars. We aimed to estimate image similarity metrics that account for BCC’s perceptual color and texture deviation from perilesional skin. In total, 176 clinical photographs of BCC were assessed by six physicians using a visual deviation scale. Internal consistency and inter-rater agreement were estimated using Cronbach’s α, weighted Gwet’s AC2, and quadratic Cohen’s kappa. The mean visual scores were used to validate a range of similarity metrics employing different color spaces, distances, and image embeddings from a pre-trained VGG16 neural network. The calculated similarities were transformed into discrete values using ordinal logistic regression models. The Bray–Curtis distance in the YIQ color model and rectified embeddings from the ‘fc6’ layer minimized the mean squared error and demonstrated strong performance in representing perceptual similarities. Box plot analysis and the Wilcoxon rank-sum test were used to visualize and compare the levels of agreement, conducted on a random validation round between the two groups: ‘Human–System’ and ‘Human–Human.’ The proposed metrics were comparable in terms of internal consistency and agreement with human raters. The findings suggest that the proposed metrics offer a robust and cost-effective approach to monitoring BCC treatment outcomes in clinical settings.

## 1. Introduction

Basal cell carcinomas (BCCs) occur in a wide range of body locations, yet their prognosis is excellent, as most BCCs do not possess aggressive biological behavior. The suggested treatment for these tumors is surgical excision [1] with adequate margins [2], which can result in a scar, however, that may significantly impact the aesthetic appearance of the patient, particularly when BCCs occur on the face. Notably, untreated BCCs also share key visual characteristics with scars, i.e., similar alterations of the skin relief (usually a protuberance) and of the local qualities of the texture and of the color of the body surface. Therefore, for the efficient management of BCCs and the evaluation of not only the effectiveness of therapeutic interventions but also of the respective long-term sequelae, it is crucial to conduct reliable assessments to better understand the visual impact of both tumors and post-treatment scars. In view of the generally non-aggressive nature of BCCs, alternative surgery methods of treatment have been developed. One of these is immunocryosurgery, i.e., the combination of imiquimod and cryosurgery in a fixed-time protocol [3]. One of the advantages of this minimally invasive approach is the reasonably good resulting scars. 

Scar assessment and response to therapy have been previously assessed with subjective scar scales [4,5]; however, in the meanwhile, noninvasive, objective, and quantitative measurement devices have been developed that seem to supersede them [6]. Technology-based scar assessment tools allow their accurate and reproducible evaluation. Lee et al. [7] reviewed objective devices for burn scar analysis classified according to the features they may assess. These included color and texture (measured by digital photographs and laser imaging); scar dimensions (using 3D photographic imaging and ultrasound); and pliability and elasticity (measured by cutometers, tissue tonometry, and elasticity probes). However, the high cost of these devices, the complexity of their usage, and the time constraints of the clinicians involved are significant obstacles to their wider adoption. Therefore, these promising, noninvasive devices are still primarily used for research and have not been incorporated into daily clinical practice [6,8]. 

On the other hand, clinical photography [9,10,11], coupled with computerized image analysis [12,13], is a cost-effective approach alternative to be used in lieu of live patient assessments to assess scars’ features. Among the different parameters characterizing a scar, color, and texture appearance are the leading parameters that contribute to the assessment of its visibility [4]. Color deviations relate to alterations of the underlying local blood perfusion rates and the concentrations of other chromophores. Texture aberrations are perceived as alterations of the smoothness, roughness, and irregularity confined to the skin surface of the scar. A previous study [12] outlined a machine-learning-assisted tool for automated burn scar severity rating (classification) based on the Vancouver Scar Scale. For implementing their multi-classifier to predict scores of the scars, the authors considered only color and texture features as input information for the classification process.

In a recent study [13], we verified the applicability of a modified scar rating system (MSRS) [8] to evaluate the impact of immunocryosurgery on the visibility of the BCC harboring skin area after treatment compared to the pretreatment visual perception of the tumor. In essence, this user-friendly scale consists of three components (‘texture’, ‘color’, and ‘height’) and utilizes patient photographs to evaluate the appearance of a specific skin area. The visibility of the target site is subjectively assessed based on increasing levels of dissimilarity when compared to the characteristics of the surrounding skin.

However, MSRS relies on subjective visual inspection. A more reliable and valid assessment of these items might help clinicians measure outcomes and develop and evaluate treatment strategies. Herein, we strike forward, breaking down visual similarity into two major sub-problems (color and texture assessments) and exploring relevant descriptors and metrics using computerized image analysis, which best predicts perceptual similarity. Although such an approach is appealing for producing transparent scoring rules, it faces immense challenges to devise features that capture the way experts perceive color and texture differences. For this, we explored a variety of similarity metrics in different color spaces, and we utilized the distances between images in the embedding space of a pre-trained deep convolutional neural network (CNN), exploiting the emergent property of deep visual presentations in predicting perceptual similarity [14]. Herewith, through analysis of BCC treatment data, we present a promising, clinical-photographs-based, robust, high-speed, and affordable computerized image analysis modality to reliably assess the visibleness of skin lesions and monitor treatment outcomes in clinical settings.

To the best of our knowledge, our work represents the initial attempt to predict perceptual similarity accurately and reliably based on clinical photography in the field of dermatology.

## 2. Materials and Methods

The use of archival photographic material for this study was approved by the Human Investigation Committee (IRB) of the University Hospital of Ioannina (approval nr.: 3/17-2-2015[θ.17)].

We used 176 photographs from 100 patients (57 males and 43 females; age range: 45–85 years) routinely treated for a facial BCC. Sixty-seven photographs were acquired from treated BCC tumors (post-treatment scars: 21 scars after standard surgical excision; 46 scars after immunocryosurgery). The remaining 109 photographs were from untreated BCC tumors at the patient’s first examination.

The data acquisition procedure consisted of two steps. In the first step, six physicians (four dermatologists and two plastic surgeons) independently graded the pre- and post-treatment BCC lesions using the scale of Mecott et al. [9]. The same photographs were then used for the subsequent second digital imaging analysis study step. More precisely, the physicians assessed the resemblance of the target, ‘lesional’, and the perilesional, ‘healthy’ skin areas on a 4-score scale of increasing color and texture visual deviation: ‘1′ (indistinguishable from the surrounding skin) ‘2’ (slight), ‘3’ (moderate), and ‘4’ (strong deviation). 

Internal consistency and interrater agreement were estimated using Cronbach’s α [14] and weighted Gwet’s AC2 [15], respectively. The mean color and texture visual deviation score was estimated and used as the “gold standard” to validate the quantitative descriptors derived from image analysis. Figure 1 demonstrates examples of averaged color and texture scores for the BCC sites.

### 2.1. Color and Texture Similarities Using Clinical Photographs

From each photo, three patches of arbitrary size were manually cropped: one patch from the target skin area $\left(S\right)$ and two sample patches (S1 and S2) from the perilesional skin area. Similarity was estimated as the mean “distance” of the target patch S from the perilesional skin patches S1 and S2:(1)$$similarity=\frac{M\left(S,S1\right)+M\left(S,S2\right)}{2}$$
where M is a “distance” metric that measures the degree of color and texture deviation of the target skin area from the surrounding healthy skin.

#### 2.1.1. Perceptual Color Similarity

To measure color similarity, the metric M (Equation (1)) operates on the patches’ mean color vectors. Humans perceive colors differently from the way colors are presented in different color spaces. Aiming to measure color similarity as perceived by human raters, we explored different color spaces (RGB, YIQ, and CIELAB) and metrics (Euclidean distance, Bray–Curtis distance, and deltaE94).

Color modeling is essential in various image-processing applications based on skin color information [16,17]. Many color spaces have been developed to represent the color information of color images. The default color space for most image-capturing and storing devices is RGB (red, green, and blue). However, in computer vision and image processing, RGB color space is converted into other color spaces through linear or non-linear transformations. 

Important color spaces successfully used in skin lesion applications are the YIQ and CIELAB color models [18,19,20]. The YIQ color model was explicitly designed to consider the non-linear response of the human eye to different colors. YIQ separates the luminance (Y) and color information (I and Q components). The I component ranges from blue to orange, and the Q component ranges from green to purple. The CIELAB color space, also referred to as L*a*b*, was intended as a pseudo-uniform color space, such that the Euclidean distance between two specified colors in this space is proportional to the color difference between these colors perceived by a standard observer. The L*a*b* color model also separates brightness L* from chromaticity components (a*b*). Chromaticity a* ranges from green to red, and chromaticity b* ranges from blue to yellow. 

Color differences were estimated using the standard Euclidean distance and the Bruy–Curtis distance (BCD). For example, in CIELAB color space, assuming the mean color vectors $C1=(L_{1},a_{1},b_{1})$ of image patch S1 and $C2=L_{2},a_{2},b_{2}$ of image patch S2 , their Euclidean distance and the BCD are given by:(2)$$deuc\left(C1,C2\right)=\sqrt{{\left(L_{1}-L_{2}\right)}^{2}+{\left(a_{1}-a_{2}\right)}^{2}+{\left(b_{1}-b_{2}\right)}^{2}}$$
(3)$$dBCD\left(C1,C1\right)=\frac{\left|L_{1}-L_{2}\right|+\left|a_{1}-a_{2}\right|+\left|b_{1}-b_{2}\right|}{L_{1}+L_{2}+a_{1}+a_{2}+b_{1}+b_{2}}$$

Likewise, we estimated color differences in YIQ and RGB color spaces.

The Euclidean distance between two color vectors in CIELAB color space (Equation (2)) is known as the DeltaE color difference. DeltaE has been successfully employed in several studies to quantify skin color differences using digital photography [21,22,23]. In the present study, we additionally explored DeltaE94, a modified formula that is proposed to represent the human perception of color differences better than DeltaE [24].

The BCD metric is often used in environmental science and biology, but recent studies have highlighted the performance of BCD in medical information [25] and medical image retrieval [26,27]. BCD is a normalized metric that treats the variations among low and high values alike, with a nice property for positive values (in our case, skin color has positive values in YIQ/L*a*b* color spaces): the BCD lies between 0 and 1, where zero means actual similarity. 

#### 2.1.2. Perceptual Texture Similarity

In computer vision, a large body of literature has been devoted to texture feature extraction to perform tasks such as image classification, segmentation, and retrieval [28]. However, in a survey study on perceptual textural similarity estimation, Dong et al. [29] demonstrated that there is no simple relationship between the perceptual attributes and the computational features of texture images. The survey concluded that features from image embeddings derived from pre-trained CNNs outperform the conventional features. The latter verified the observations of Zhang et al. [30], who first revealed that perceptual similarity is an emergent property shared across deep visual presentations. In a subsequent study, Gao et al. [31] proposed a framework to predict fine-grained perceptual texture similarity by combining layer-wise deep feature similarity and similarity between images’ contour maps.

Motivated by the studies above, we used the VGG16 [32] network pre-trained on ImageNet [33] for feature extraction and texture similarity estimation. Consequently, to measure texture differences, the metric M (Equation (1)) operates on the deep representations of the patches obtained through the VGG16 network.

CNNs process images by convolving multiple filters over the input image to extract local patterns and features. The output of each filter is a two-dimensional feature map that captures the filter’s response at each spatial location of the input image. A convolutional layer with N filters (channels) generates N feature maps. Subsequent convolutional layers combine these feature maps to form higher-level features that capture increasingly complex visual patterns.

Assuming a pair of image patches S1 and S2, we denote their pair of activations as:(4)$$F=<f_{1}^{L},f_{2}^{L}>$$

$f_{1}^{L}$ and $f_{2}^{L}$ are the activations of image patches S1 and S2 from layer L, respectively.

The cosine similarity estimates the texture similarity of pair *F* [31]. Considering final image embeddings from the fully connected layers, the cosine distance is estimated between the corresponding deep feature vectors. 

The cosine similarity between feature maps is estimated as follows: For each spatial position, a vector with a length equaling the number of filters in the $L^{th}\;$ layer exists. For the same spatial position of $f_{1}^{L}$ and $f_{2}^{L}$ we calculate the cosine similarity of the two vectors. The final similarity in the Lth layer is the average similarity across the spatial positions. The expressive ability of different layers of VGG16 was tested for the texture similarity calculation.

### 2.2. Validation of Similarity Metrics 

To validate how accurately our metrics predict the perceptual color and texture scores, we transformed the calculated similarities into discrete values from 1 to 4 using ordinal logistic regression (OLREG) models. We randomly split the data set into training and validation sets. The model construction uses the training set and the validation set to calculate its accuracy. As we aimed to predict perceptual similarity as accurately as possible, to select from different options for calculated similarity metrics, we used the mean squared error (MSE), which is defined as:(5)$$MSE=\frac{1}{N}\sum {\left(y-\hat{y}\right)}^{2}$$
where y and $\hat{y}$ are the mean perceptual and predicted (automated) scores, respectively, and N is the number of validated scores. We repeated the random splitting process multiple times to prevent biases and provide more reliable and robust estimates.

Best similarity metrics minimized the mean MSE  error ($\bar{MSE}$); for these metrics, mean absolute accuracy $\bar{ACC}\;$ and mean adjacent accuracy $\;\bar{ACC\_adj}$ were also reported. Absolute accuracy refers to the exact agreement between predicted and perceptual scores. Adjacent accuracy refers to the adjacent agreement where predicted and perceptual scores do not differ more than by one level. With this method, we identified the ‘automated’ similarity metrics that best predicted the physicians’ ‘human’ perceptual scores. 

We employed K-means cluster analysis to highlight how the chosen similarity metrics effectively capture perceptual similarities in a meaningful manner. 

#### “Human” versus Automated Score Agreement Compared to between Experts Agreement

To further examine the performance of the ‘automated’ image similarity metrics, we compared the level of agreement between the raters’ and automatically generated scores (‘Human–System’ agreement) versus the level of agreement between scores assigned by the different expert raters (‘Human–Human’ agreement). We aimed to determine whether our proposed framework has the potential to perform at a level similar to that of an expert. Consistency and agreement assessments were conducted on a random validation round, using Cronbach’s α and quadratic Cohen’s kappa [14]. We utilized a box plot analysis to visualize the levels of agreement between the two groups: ‘Human–System’ (‘H-S’) and ‘Human–Human’ (’H-H’).

## 3. Results

All experts yielded ‘excellent’ consistency (α > 0.9) with at least ‘good’ reliability (AC2 > 0.70) for both texture and color assessments. For each target skin area (BCC or post-treatment scar), the rounded mean score was calculated and used to validate the color and texture similarity metrics. Table 1 presents the distribution of mean scores for color and texture deviations.

To ensure a balanced representation of the four scales during the training of the OLREG models, the training set consisted of 20 samples randomly selected from each scale in each split. (Training set: N = 80; validation set: N = 96). Different color and texture models were validated in terms of MSE over one hundred repetitions. 

Considering the color similarity, the metric that minimized $\bar{MSE}$ was BCD in the YIQ model ($\bar{MSE}$ = 0.564, 95% CI: 0.560–0.567), which also yielded mean absolute accuracy $\bar{ACC}=$0.543 (95% CI: 0.541–0.545) and mean adjacent accuracy $\bar{ACC\_adj}=$ 0.964 (95% CI: 0.963–0.965). Figure 2 depicts the $\bar{MSE}$ performance for different color spaces and distance metrics. 

Perceptual texture scores were best predicted by employing deep representations from the rectified layers ‘relu6′ ($\bar{MSE}$ = 0.512, 95% CI: 0.509–0.515), with absolute accuracy $\bar{ACC}=$ 0.538 (95% CI: 0.535–0.540) and mean adjacent accuracy $\bar{ACC\_adj}=$ 0.983 (95% CI: 0.982–0.984).

Figure 3 summarizes the performance of different layers of VGG16 in predicting perceptual scores.

Figure 4 presents a qualitative k-means cluster analysis with k = 4 using color (BCD-YIQ) and texture (relu6) similarity metrics. For each cluster, we estimated the mean perceptual score for color and texture similarity. The similarity metrics successfully cluster the perceptual scores in a consistent manner, that is, items within the same cluster have similar perceptual scores for both color and texture similarity. For example, the first cluster gathers BCC sites that are perceptually similar to healthy skin.

### Comparing the ‘between Humans’ Agreement with That between Humans and the System

For a random run using the best-resulted metrics (color: ‘BCD-YIQ’; ‘texture: relu6′), we acquired the color and texture similarity predictions for the validation set (N = 96). Assuming that automated scores were produced from a seventh rater, we analyzed the consistency of ratings among seven raters using Cronbach’s alpha coefficient. The initial Cronbach’s alpha coefficient for the ratings was 0.937 for color and 0.958 for texture, indicating an excellent level of internal consistency. A rater removal analysis was conducted to further investigate the impact of individual raters on the overall consistency. Each rater was removed one at a time, and Cronbach’s alpha was recalculated for the remaining six raters. Upon rater removal, Cronbach’s alpha coefficient ranged from 0.917 to 0.934 for color and 0.950 to 0.953 for texture, suggesting that the presence of systems predictions did not influence the overall consistency of the ratings.

We further assessed the agreement using Cohen’s kappa for each pair of raters. The mean agreement for all 15 rater combinations (taking all combinations of six by two) was 0.65 (SD = 0.1) and 0.76 (SD = 0.03) for the color and texture scores, respectively. Likewise, the mean agreement resulting from the six machine–human pairs was 0.64 (SD = 0.04) and 0.75 (SD = 0.02) for color and texture, respectively. Figure 5 depicts a box plot analysis comparing the groups of human–human agreements (N = 15 pair agreements) and the human–system agreement (N = 6 pair agreements) regarding the color and texture appearance.

Overall, the distributions of Cohen’s kappa values for the groups ‘S-H’ and ‘H-H’ are quite similar, suggesting comparable central tendencies among the human–human and system–human agreement for both color and texture scores (*p* > 0.05; Wilcoxon rank-sum test). 

The analysis of perceptual agreement between humans for color (‘Color H-H’) and texture (Texture ‘H-H’) revealed that perceptual texture agreement was higher than color perceptual agreement (*p* < 0.01; Wilcoxon rank-sum test). 

## 4. Discussion

BCCs can occur in a wide range of body locations, but BCCs of the face are especially problematic, so prioritizing the aesthetic outcomes of treatment is highly valued by patients [34]. In order to ensure effective decision-making in managing BCCs and evaluating therapeutic interventions over time, accurate assessments are essential. These assessments should provide insights into the visual characteristics of tumors and post-treatment scars. In the present study, our objective was to build upon our previous findings [13] and develop an automated tool that can quantify observed changes in pre- and post-treatment BCC sites and simulate the way experienced clinicians assess the visual changes induced in the affected skin sites.

Given that color and texture are critical attributes that decisively determine the clinical appearance of the skin lesions, our focus was on quantifying deviations of these characteristics from the surrounding healthy skin through image analysis and proposing similarity models that are congruent with the way experts make their judgments.

In the context of color similarity, we tested widely accepted approaches, including not only L_2_ Euclidean distance but also BCD as an alternative to RGB color spaces (YIQ/L*a*b*). Interestingly, our results revealed outstanding performance by the BCD metric in the tested color spaces. Better normalization and goodness of fit to the ordinal proximity data are the foremost benefits of BCD, also verified by previous studies in medical information and image retrieval.

Considering perceptual texture similarity, our study builds upon and contributes to the existing body of research in this area that involves the use of internal activations of deep convolutional networks trained for large-scale image classification tasks to measure perceptual similarity. In general, CNNs consist of convolutional and fully connected layers. Convolutional layers learn progressively from fine to large spatial extend image representations, whereas the top fully connected layers learn to capture image-wide global information. Our experiments indicated that rectified embeddings from the fully connected layer ‘fc6′ (relu6 layer) are the most predictive for target skin texture similarity—a result that agrees with the intuition behind deep embeddings and visual perception. Moreover, the ReLU (rectified linear unit) layer is a key component of the VGG16 architecture that sets all negative values to zero while leaving positive values unchanged and is responsible for capturing and emphasizing relevant patterns and information while suppressing less important or irrelevant features. 

The proposed similarity metrics produce scores that, in terms of internal consistency and agreement, are comparable with those obtained from human raters. This finding aligns with and supports our aim of devising valid computational metrics to predict perceptual color and texture similarity. Moreover, estimated metrics enable a quantitative analysis of skin properties that were previously reliant on subjective descriptions. Quantifying these properties is crucial because discrete categories (‘1’,’2’,’3’,’4’) struggle to capture subtle changes in skin appearance. K-means cluster analysis not only highlights the effectiveness of the selected similarity metrics to represent perceptual similarities in a meaningful way but also provides evidence of subtle discrimination of perceptually similar cases in the continuous similarity space (Figure 4). Our continuous similarity metrics can provide more informative measurements, allowing for increased sensitivity in detecting and tracking changes in BCC sites over time.

In the box plot analysis (Figure 5), the median values for texture perception agreement were noticeably higher, indicating stronger consensus among the human raters. On the other hand, color perception showed relatively more variability, as evidenced by the wider spread of the box plot. These findings suggest that experts tend to have higher agreement levels when evaluating texture than color appearance. It is noteworthy that the same relationships between color and texture perception as found in the human-to-human comparisons are also evident in the pairwise human-to-machine evaluations: a higher degree of agreement in the evaluation of texture compared to color, while at the same time, a proportionally higher degree of dispersion of Cohen’s kappa estimates in the case of color evaluation. Taken together, these latter observations support our hypothesis that the currently configured “machine” behaves similarly to a “human evaluator” when it comes to assessing the visual similarity of selected confined skin lesions from their surrounding skin. 

An important factor that affects texture quantification is the image scale. In clinical photography, the resolution specifications of the camera and also the distance of the patient from the camera affect the image scale. In our study, we used clinical photographs, retrieved from dermatology and plastic surgery clinic archives. Moreover, these photographs were acquired by different operators (GG and SK) and using cameras with different spatial resolutions (4016 × 6016; 3648 × 2736; 768 × 1024). Using an internal marker (fiducial marker), the image scale was estimated to range from about ~10 to 45 pixels/mm.

Likewise, a human rater—when observing a clinical photograph and evaluating the texture and color differences between a target skin area and the surrounding healthy skin—demonstrates a certain level of tolerance against the image scale; the estimated similarity scores show a similar level of invariance or robustness against the image scale. This tolerance is inherent to the method itself that compares regions or patches from the same photograph and focuses on the relative differences between the target area and its surrounding regions.

Moreover, considering the estimated texture similarity, image representations learned by deep convolutional neural networks often exhibit tolerance against image scale variations. This means that the representations learned by these models can generalize well across different scales of input images. CNNs commonly consist of multiple convolutional layers followed by pooling layers. Convolutional layers employ filters that detect local patterns in the input images. Pooling layers, typically used after convolutional layers, downsample the spatial dimensions of the feature maps while preserving their essential information. This property allows the network to recognize patterns or features regardless of their size in the input image.

However, one of the main limitations of our study is the careful selection of image patches, which can potentially impact the accurate estimation of similarity. Particularly for color similarity, it is crucial to choose a rectangular area within the target patch that includes a minimal amount of perilesional skin. Also, there is a need to improve perceptual color similarity metrics and enable better quantification of color relationships based on human perception. In addition, the inclusion of shiny skin patches due to light reflections affects the estimated metrics. The latter could be effectively addressed by using cross-polarized photography as suggested and verified by previous researchers [35,36]. 

In the future, efforts should be made to expand the BCC image dataset, including the judgments of experts, and ensure that the dataset captures a wide range of variations and characteristics associated with BCCs [37,38]. 

## 5. Conclusions

Overall, the reported experimental results promise a robust, high-speed, and affordable modality to monitor BCC treatment outcomes in clinical settings. The proposed metrics allow for quantified examination of a skin area which is of great importance towards reliable skin visibleness assessments. Moreover, our approach to model color and texture lesion similarity from the surrounding healthy skin is a promising paradigm for the further development of a valid and reliable scar assessment tool.

## Figures and Tables

**Figure 1 cancers-15-03539-f001:**
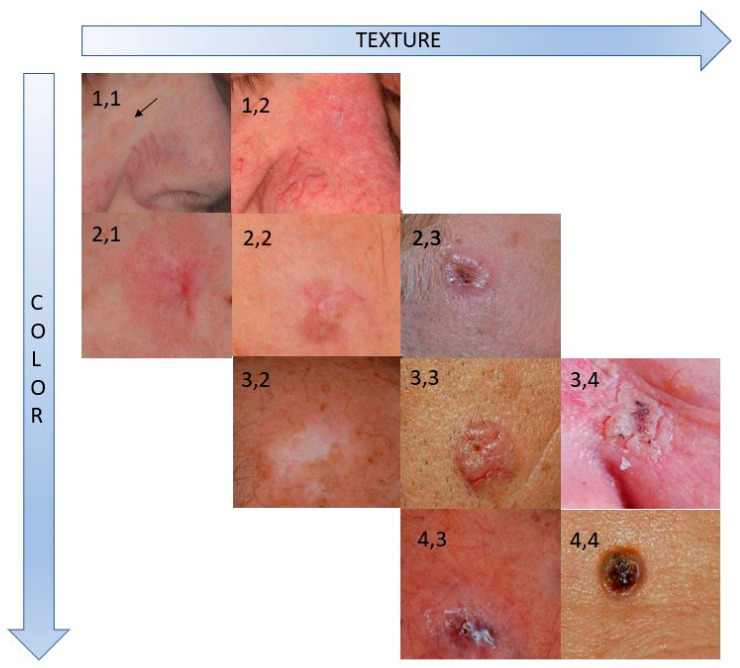
BCC target sites rated (mean score) for color and texture deviations from the surrounding skin. Rates are noted in the form (color, texture).

**Figure 2 cancers-15-03539-f002:**
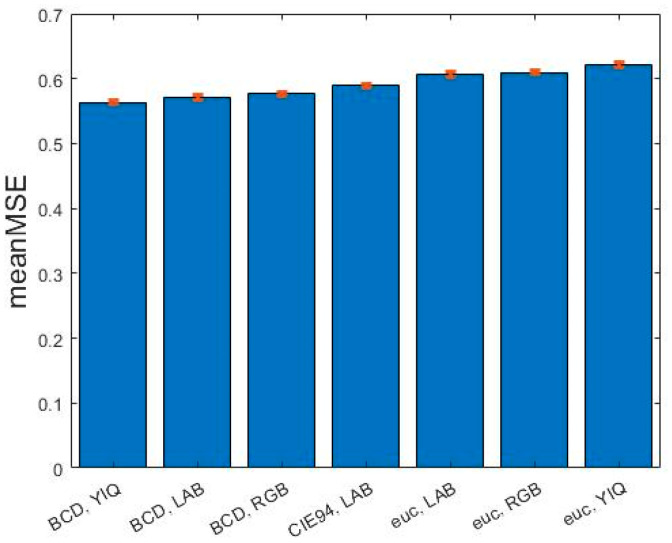
Color metrics rated with the increasing mean mean square errors ($\bar{MSE}$; red bars: 95% CI) values.

**Figure 3 cancers-15-03539-f003:**
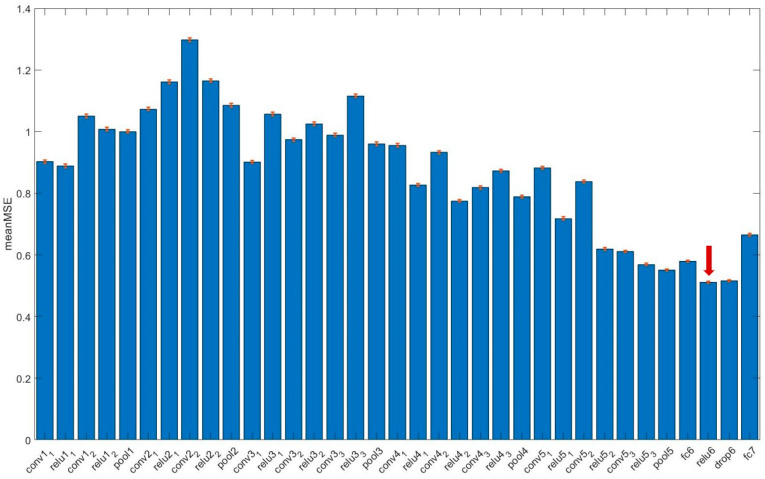
Layer-wise performance (mean MSE, mean square error, with the corresponding 95% CI as red bars) in predicting perceptual texture scores. Layer ‘relu6′ (arrow) was the most accurate according to the minimization of the mean MSE.

**Figure 4 cancers-15-03539-f004:**
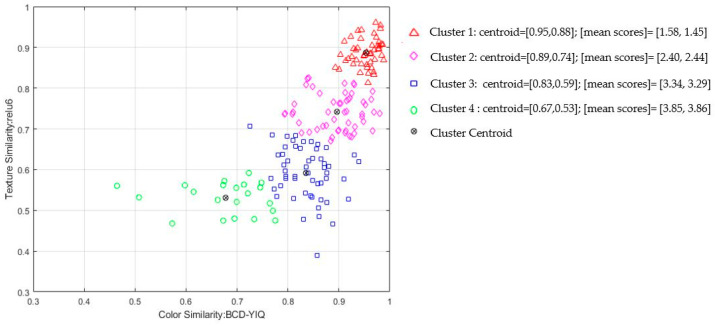
Qualitative k-means cluster analysis with k = 4 using color (BCD-YIQ) and texture (relu6) similarity metrics. The similarity metrics cluster the perceptual scores consistenty. Note that color similarity is estimated as 1-BCD, so higher values indicate greater color similarity levels.

**Figure 5 cancers-15-03539-f005:**
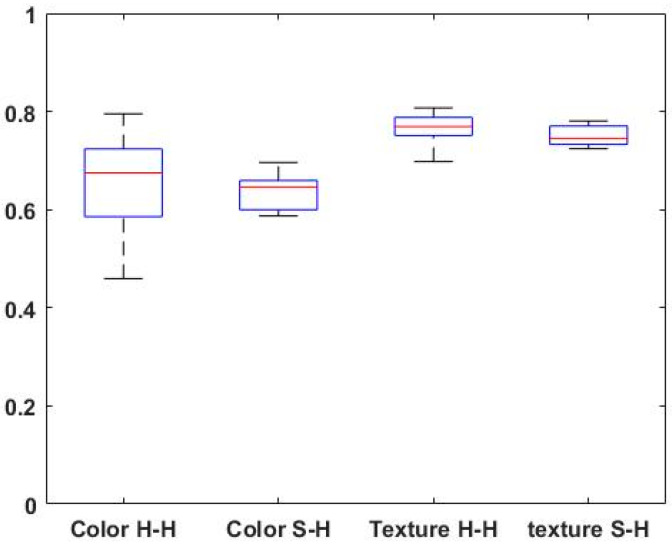
Box plot analysis of the agreement groups (pairwise Cohen’s kappa): between two human raters (‘H-H’ group) and the humans to system comparisons (‘S-H’ group) for rating color and texture similarity. The samples’ medians are shown in red.

**Table 1 cancers-15-03539-t001:** Mean color and texture deviation scores of the pre- and post-treatment BCC tumors.

	Color				Texture			
	‘1’	‘2’	‘3’	‘4’	‘1’	‘2’	‘3’	‘4’
BCC tumors	0	28	36	45	0	24	46	39
Post-treatment scars	24	33	8	2	30	29	5	3
Total	24	61	44	47	30	53	51	42

## Data Availability

The data presented in this paper are not publicly available at this time but may be obtained from the authors upon request.

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
