# Peer review of "Image Perceptual Similarity Metrics for the Assessment of Basal Cell Carcinoma"

_cancers, 2023, doi:10.3390/cancers15143539_

Round 1

Reviewer 1 Report

Dear Authors

I have enjoyed reading your paper. I found it to be an interesting proposal. I have a minor comment online 48. I would not say that BCC " do not possess an aggressive biological behavior" as a few might do. I would change to " most ...do not possess an aggressive biological b ..."

Author Response

Comment: I have a minor comment online 48. I would not say that BCC " do not possess an aggressive biological behavior" as a few might do. I would change to " most ...do not possess an aggressive biological b ..."

Answer: The comment has been addressed in the first paragraph of Introduction:

 “Basal cell carcinomas (BCC) occur primarily on the face, yet their prognosis is excellent, as most BCC do not possess an aggressive biological behavior.”

We thank the reviewer for his supportive comment.

Reviewer 2 Report

The authors perform a study about Ιmage perceptual similarity metrics for the assessment of basal 2 cell carcinoma, although the study is of interest some changes are needed.

Please group Table 1 and Table 1 in only 1 single Table

You should stress in the discussion what is the main clinical relevance of your study. 

In your study you did not discuss the aggressive variants of BCC, which can pose significant diagnostic and therapeutic challange to clinicians and surgeons. In this regard please read and add this recent article "Clinical and Dermoscopic Factors for the Identification of Aggressive Histologic Subtypes of Basal Cell Carcinoma. PMID: 33680953; PMCID: PMC7933517."

Please, in this regard, specify how your method can be affected by aggressive variants of BCC.

I will be happy the review the revision of the article.

Thank you.

Author Response

Comment 1: Please group Table 1 and Table 2 in only 1 single Table”

Answer: The two tables have been merged as suggested, line 255 in the revised Manuscript.

Comment 2: “You should stress in the discussion what is the main clinical relevance of your study”

Answer: In Introduction we have expanded the first paragraph (line 320) to declare the clinical relevance of our study as suggested:

“BCC is a tumor that primarily occurs on the face, and prioritizing the aesthetic outcome of the treatment is highly valued by patients, compared to the treatment's effectiveness in terms of cure and recurrence rates [34]. In order to ensure effective decision-making in managing BCC and evaluating therapeutic interventions over time, accurate assessments are essential. These assessments should provide insights into the visual characteristics of tumors and post-treatment scars. In the present study, our objective was to build upon our previous findings [13] and develop an automated tool that can quantify observed changes in pre- and post-treatment BCC sites and simulate the way experienced clinicians assess the visual changes induced in the affected skin sites”.

Comment 3: In your study you did not discuss the aggressive variants of BCC, which can pose significant diagnostic and therapeutic challange to clinicians and surgeons. In this regard, please read and add this recent article "Clinical and Dermoscopic Factors for the Identification of Aggressive Histologic Subtypes of Basal Cell Carcinoma. PMID: 33680953; PMCID: PMC7933517. Please, in this regard, specify how your method can be affected by aggressive variants of BCC”.

Answer: We agree that the recognition/approach of the "aggressive variants" of BCC, which are particularly common among neoplasms affecting the facial area, as in the present study, is a significant clinical challenge. However, the study of these variants as a specific subgroup was beyond the scope of the current study, which aims to identify (and simulate/replicate) specific visual characteristics of selected visually recognizable morphological deviations in the superficial skin structure. Testing the applicability of the tools in processing indicative characteristics of different clinical entities is among the future goals of the project. In this context, the comparative analysis of "aggressive variants" of BCC is one of our priorities. Thank you for your interesting reference, which we have included in Revised Manuscript (ref 37) discussing our future plans (Discussion; last paragraph):

“In the future, efforts should be made to expand the BCC image dataset, including the judgments of experts, and ensure that the dataset captures a wide range of variations and characteristics associated with BCC [37],[38].  

We thank the reviewer for his valuable and supportive comments

Reviewer 3 Report

Overall an interesting paper about a topic with a big future.

The Authors should be commended for the very good analysis of the data.

There are a number of issues that need addressing to improve the clinical application and relevance of this study:

1 Page 48 states BCC is not aggressive. BCC is not one uniform group of malignant tumours. Many common BCC subtypes are aggressive. These several subtypes of BCC can have aggressive behaviour such as deep and wide destruction of host tissue, invade nerves (perineural invasion) and in very rare cases result in a BCC killing the patient. The Authors need to mention aggressive subtype BCC and what morbidity and mortality they can cause. There are also published papers on the appearance of aggressive BCC and how they differ in clinical and dermatoscopic features (which includes colours) compared to other BCC subtypes. Variation in the anatomic location of BCC have also been published showing BCC on the face are often aggressive subtypes (see paper by Pyne, Barr and Myint on over 4,000 BCC in Dermatology Practical and Conceptual 2018). 

2 After BCC is treated by surgery (or other modalities) the site changes colour and appearance over time. The time between treatment and the study data collection should be discussed as this will affect the results. 

3 The depth resolution limits of examining skin and effect of magnification/image size and quality can affect results. Discuss further??

Suggestions:

Page 20 Change "device" to "devise".

Page 29 Change "We aimed to estimate" to "We quantified".

Pages 40 and 41 Change "proposed" to "stated".

Page 48 Rewrite to clarify/define aggressive subtypes of BCC  

Page 94 Change "we strike forward" to something less tabloid.

Author Response

Comment 1: “Page 48 states BCC is not aggressive. BCC is not one uniform group of malignant tumours. Many common BCC subtypes are aggressive. These several subtypes of BCC can have aggressive behaviour such as deep and wide destruction of host tissue, invade nerves (perineural invasion) and in very rare cases result in a BCC killing the patient. The Authors need to mention aggressive subtype BCC and what morbidity and mortality they can cause. There are also published papers on the appearance of aggressive BCC and how they differ in clinical and dermatoscopic features (which includes colours) compared to other BCC subtypes. Variation in the anatomic location of BCC have also been published showing BCC on the face are often aggressive subtypes (see paper by Pyne, Barr and Myint on over 4,000 BCC in Dermatology Practical and Conceptual 2018).”

Answer: We have corrected that not all, but most BCCs do not possess an aggressive biological behavior as suggested by the first Reviewer (see Reviewer 1; comment)

Thank you for your suggestion, your interesting reference, has been included in the Revised Manuscript (ref 38) discussing our future plans (Discussion; last paragraph):

“In the future, efforts should be made to expand the BCC image dataset, including the judgments of experts, and ensure that the dataset captures a wide range of variations and characteristics associated with BCC [37],[38].”

Comment 2: After BCC is treated by surgery (or other modalities) the site changes color and appearance over time. The time between treatment and the study data collection should be discussed as this will affect the results”.

Answer:  This statement is true, when comparing the treatment outcomes from different therapeutic interventions (i.e comparing surgical excisions and immunocryosurgery).  However, in this work we aimed to develop an automated tool able to quantify observed changes in pre- and post-treatment BCC sites and simulate the way experienced clinicians assess the visual changes induced in the affected skin sites. Having an accurate and reliable tool for assessing BCC sites, the next step is to employ this tool to monitor treatment outcomes and/or compare between therapeutic modalities.

Comment 3: “The depth resolution limits of examining skin and effect of magnification/image size and quality can affect results. Discuss further??”

Answer: Thank you for your comment that gave us the chance to discuss the tolerance of our method across different image scales and indicate improvements for future developments. In discussion (line 377) we have added the following text:

“An important factor that affects the texture quantification is the image scale. In clinical photography, the resolution specifications of the camera but also the distance of patient from camera affect the image scale. In our study we have used clinical photographs, retrieved from the archives of dermatology and surgery clinics. Moreover, these photographs were acquired by different operators (GG, SK), and using cameras with different spatial resolution (4016x6016;3648x2736;768x1024). Using an internal marker (fiducial marker), the image scale was estimated to range from about ~10 to 45 pixels/mm.

Likewise, a human rater-when observing a clinical photograph and evaluating the texture and color differences between a target skin area and the surrounding healthy skin- demonstrates a certain level of tolerance against image scale, the estimated similarity scores show a similar level of invariance or robustness against image scale. This tolerance is inherent to the method itself that compares regions or patches from the same photograph and focus on the relative differences between the target area and its surrounding regions.

Moreover, considering the estimated texture similarity, image representations learned by deep convolutional neural networks often exhibit tolerance against image scale variations. This means that the representations learned by these models can generalize well across different scales of input images. CNNs commonly consist of multiple convolutional layers followed by pooling layers. Convolutional layers employ filters that detect local patterns in the input images. Pooling layers, typically used after convolutional layers, down sample the spatial dimensions of the feature maps while preserving their essential information. This property allows the network to recognize patterns or features regardless of their size in the input image.

However, one of the main limitations of our study is the careful selection of image patches, which can potentially impact the accurate estimation of similarity. Particularly for color similarity, it is crucial to choose a rectangular area within the target patch that includes the minimal amount of perilesional skin.  Also, there is a need to improve perceptual color similarity metrics and enable better quantification of color relationships based on human perception. In addition, the inclusion of shiny skin patches due to light reflections affects the estimated metrics. The latter could be effectively addressed by using cross polarized photography as suggested and verified by previous researchers [34],[35]”

We thank the reviewer for his valuable and supportive comments